# Dietary sodium, potassium, and cardiometabolic risk: A cross-sectional analysis of hypertension in U.S. adults from NHANES 2017–2018

**Xiao Luo, Xuan Zhang, Longfeng Ran, Zhuojun Kang, Xinyu Fu, Xin Mu***

Department of Neurology, Chengdu Integrated TCM & Western Medicine Hospital, Chengdu University of Tranditional Chinese Medicine, Chengdu, China

* 1378570@qq.com

## Abstract

### Background

The relationships between dietary sodium and potassium intake and hypertension remain controversial, with recent population-based analyses yielding inconsistent findings. This study aimed to evaluate the associations of sodium and potassium intake with hypertension among U.S. adults and to explore potential interactions with demographic and lifestyle factors.

### Methods

We conducted a cross-sectional analysis using data from 5,569 adults aged ≥20 years in the NHANES 2017–2018 cycle. Dietary intake was assessed using 24-hour dietary recall. Hypertension was defined based on self-reported diagnosis. Multivariable logistic regression models were used to estimate adjusted odds ratios (ORs) and 95% confidence intervals (CIs). Restricted cubic spline models examined nonlinear trends, and subgroup analyses were stratified by sex, age, and BMI.

### Results

No significant associations were found between sodium or potassium intake and the odds of hypertension after adjusting for covariates (OR ≈ 1.00, P > 0.05) and findings were consistent when including the Na/K ratio. Results remained stable across sensitivity models and spline analysis. A non-significant inverse trend was observed for dietary fiber intake. Subgroup analyses suggested slightly stronger associations among older adults and individuals with obesity, although interaction terms were not statistically significant.

**Data availability statement:** All data used in this study are publicly available from the National Health and Nutrition Examination Survey (NHANES) at https://www.cdc.gov/nchs/nhanes/. No additional supporting files are required because the dataset is openly accessible without restriction.

**Funding:** This work was supported by the Joint fund of Chengdu Health Commission and Chengdu University of Traditional Chinese Medicine (XM, grant number WXLH202406002), (XL, grant number WXLH202403263). The funders had no role in study design, data collection and analysis, decision to publish, or preparation of the manuscript.

**Competing interests:** I have read the journal's policy and the authors of this manuscript have the following competing interests:The authors declare no competing interests.

## Conclusion

In this nationally representative sample, sodium and potassium intake were not independently associated with hypertension risk. These findings highlight the complexity of diet–blood pressure relationships and the importance of considering broader dietary patterns and individual characteristics in hypertension prevention strategies.

## Introduction

Hypertension remains a leading global public health concern, affecting over 1.28 billion adults worldwide and contributing significantly to cardiovascular morbidity and mortality [1,2]. It is a major modifiable risk factor for stroke, heart failure, renal disease, and premature death [3–5]. In the United States, nearly half of adults have elevated blood pressure, yet many remain undiagnosed or inadequately controlled [6]. Given its high prevalence and substantial health burden, effective prevention and management of hypertension are critical public health priorities.

Dietary factors play a pivotal role in the development and control of hypertension. Among them, sodium and potassium intake have received considerable attention due to their physiological effects on vascular tone and fluid balance [7,8]. Excessive sodium intake is widely recognized to increase blood pressure by promoting water retention and vascular stiffness, while potassium may lower blood pressure by enhancing natriuresis and vasodilation [9]. Accordingly, global health organizations such as the World Health Organization (WHO) and the American Heart Association (AHA) recommend reducing sodium intake and increasing potassium consumption as key strategies to prevent and manage hypertension [10,11].

Numerous epidemiological studies and clinical trials have demonstrated a strong relationship between dietary sodium and potassium intake and blood pressure regulation [12–14]. Landmark studies such as the INTERSALT study and the DASH-Sodium trial have provided compelling evidence that high sodium intake is positively associated with elevated blood pressure, whereas higher potassium intake is inversely associated with hypertension risk [15]. These findings have formed the foundation for current dietary guidelines, which emphasize sodium reduction and potassium enhancement as central components of hypertension prevention strategies.

However, recent analyses have raised questions about the consistency and generalizability of these associations. Some cross-sectional studies based on large-scale population datasets, including previous the National Health and Nutrition Examination Survey (NHANES) cycles, have reported weak or nonsignificant associations between sodium or potassium intake and hypertension after adjusting for potential confounders [16–18]. Several factors may contribute to these discrepancies, including differences in dietary assessment methods (e.g., 24-hour recall vs. urinary excretion), variations in population characteristics, and the role of unmeasured lifestyle or genetic factors. Moreover, the individual effects of sodium and potassium may be influenced by other dietary components, such as fiber, cholesterol, or total energy

intake, and by population-specific susceptibilities, such as age, sex, or body mass index (BMI). These inconsistencies underscore the need for updated, comprehensive analyses using recent population data and robust statistical approaches to clarify the relationships between dietary intake and hypertension risk in contemporary populations.

In light of these uncertainties, we conducted a cross-sectional study using data from the 2017–2018 cycle of the NHANES to investigate the associations between sodium and potassium intake and hypertension in a nationally representative sample of U.S. adults. In addition to examining the primary relationships, we incorporated extended dietary variables—including total energy, dietary fiber, and cholesterol intake—to provide a broader nutritional context. We further explored potential nonlinear dose–response relationships using restricted cubic spline (RCS) models and conducted stratified subgroup analyses by gender, age, and BMI to assess effect modification across population subgroups.

Our study aims to offer a nuanced and updated perspective on the role of dietary factors in hypertension, leveraging rigorous multivariable modeling, sensitivity analyses, and comprehensive dietary data. The findings may help inform future nutritional recommendations and public health strategies aimed at hypertension prevention, particularly in identifying whether sodium and potassium should continue to be prioritized as individual targets or considered in conjunction with broader dietary patterns.

## Materials and methods

### Data source and study population

This study utilized publicly available data from the NHANES 2017–2018 cycle, conducted by the U.S. Centers for Disease Control and Prevention (CDC). NHANES employs a complex, multistage, probability sampling design to obtain a nationally representative sample of the non-institutionalized U.S. population. The survey combines interviews, physical examinations, and laboratory tests to assess the health and nutritional status of adults and children.

For this analysis, we included participants aged 20 years and older with complete information on blood pressure, dietary intake, and key covariates. Pregnant individuals and participants with missing data on hypertension status, sodium or potassium intake, or other essential variables were excluded. After applying these criteria, a total of 5,569 participants were included in the final analytic sample.

### Variable definition and measurement

Hypertension status was defined based on self-reported diagnosis in the Medical Conditions Questionnaire (MCQ). Participants were classified as having hypertension if they answered "Yes" to the question: "Have you ever been told by a doctor or health professional that you have hypertension, also called high blood pressure?" Those who answered "No" were categorized as non-hypertensive. Individuals with missing responses were excluded. Hypertension was defined based on participants' self-reported physician diagnosis, consistent with previous NHANES-based analyses. We acknowledge that NHANES also provides measured blood pressure data and antihypertensive medication information, which could allow for guideline-based definitions (e.g., ≥ 130/80 mmHg or current medication use). This approach may be considered in future analyses to further validate our findings.

Dietary intake variables, including sodium (DR1TSODI) and potassium (DR1TPOTA), were obtained from the Day 1 24-hour dietary recall interview. Additional dietary variables such as total energy (DR1TKCAL), dietary fiber (DR1TFIBE), and cholesterol intake (DR1TCHOL) were also extracted for expanded analysis. All dietary intake variables were standardized (z-scores) before inclusion in regression models to facilitate comparison across different units and scales.

Covariates were selected based on prior literature and biological plausibility. Demographic variables included age (RIDAGEYR), gender (RIAGENDR), and race/ethnicity (RIDRETH1, categorized as Non-Hispanic White, Non-Hispanic Black, Mexican American, Other Hispanic, and Other Race including multiracial). Socioeconomic status was assessed by education level (DMDEDUC2, categorized into six levels) and marital status (DMDMARTL). Lifestyle factors included smoking status

(SMQ020, categorized as smoker or non-smoker) and alcohol consumption (ALQ101, categorized as yes/no). Physical activity was derived from PAQ605 and PAQ620, indicating engagement in moderate or vigorous recreational activity.

## Dietary assessment

Dietary intake data were collected through the 24-hour dietary recall interview conducted in-person at the Mobile Examination Center (MEC) using the USDA's Automated Multiple-Pass Method (AMPM). For this study, only the Day 1 dietary recall was used to ensure uniformity across participants and to avoid potential bias from incomplete second-day data.

Sodium intake (mg/day) and potassium intake (mg/day) were extracted from the total nutrient intake file (DR1TOT), representing the total amount consumed from all food and beverage sources during the 24-hour period prior to the interview. In addition to sodium and potassium, the following dietary variables were included in extended analyses: total energy intake (DR1TKCAL, kcal/day), dietary fiber intake (DR1TFIBE, g/day), and cholesterol intake (DR1TCHOL, mg/day). All dietary variables were analyzed both as continuous variables and in standardized (z-score) form to facilitate comparison across models.

Although the 24-hour dietary recall is a validated and widely applied method in NHANES, it is inherently subject to recall bias and day-to-day variation, particularly for sodium intake due to hidden or unreported sources such as table salt and processed foods. Urinary biomarkers, which can provide more objective estimates of sodium and potassium exposure, were not used in this study because they are available only for a limited subset of participants in specific NHANES cycles, resulting in reduced statistical power and representativeness. Therefore, dietary recall data were used to ensure consistency and comparability across the full sample.

## Statistical analysis

All statistical analyses were performed using R software (version 4.2.0). Descriptive statistics were used to summarize participant characteristics by hypertension status. Continuous variables were presented as means with standard deviations (SD), and categorical variables as counts and percentages. Differences between groups were assessed using Student's t-test for continuous variables and chi-square tests for categorical variables.

To examine the associations between dietary factors and hypertension, we constructed multivariable logistic regression models. Four models were sequentially adjusted

Model 1: unadjusted.

Model 2: adjusted for age and gender.

Model 3: additionally adjusted for race/ethnicity, education level, and marital status.

Model 4: further adjusted for smoking status, alcohol consumption, and physical activity.

All dietary variables were standardized using z-scores prior to entry into regression models. Odds ratios (ORs) with 95% confidence intervals (CIs) were reported. A two-sided p-value < 0.05 was considered statistically significant.

RCS models were used to explore potential nonlinear dose–response relationships between sodium and potassium intake and the odds of hypertension. Splines with 4 knots were placed at the 5th, 35th, 65th, and 95th percentiles of the intake distributions. Models were adjusted for the same covariates as Model 4.

Subgroup analysees were performed by stratifying the sample based on gender (male/female), age (<60 vs. ≥ 60 years), and body mass index (BMI, normal vs. overweight/obese). Interaction terms were tested to evaluate potential effect modification. Finally, to explore associations between dietary intake and continuous blood pressure measures, we conducted linear regression analyses separately for systolic blood pressure (SBP) and diastolic blood pressure (DBP), stratified by hypertension status.

## Ethics statement

NHANES protocols were approved by the NCHS Research Ethics Review Board, and all participants provided informed consent. This secondary analysis of de-identified public data was exempt from additional institutional review.

## Results

### Baseline characteristics of participants

A total of 5,569 participants were included in this analysis, comprising 3,874 individuals without hypertension and 1,695 with hypertension. The mean age of the study population was 59.7 ± 12.4 years, with hypertensive participants being substantially older (62.9 ± 13.5 years) than their non-hypertensive counterparts (46.5 ± 17.2 years, P<0.001).

Regarding gender distribution, males accounted for 51.0% of the non-hypertensive group and 42.8% of the hypertensive group (P<0.001). Significant racial and ethnic differences were observed: Mexican Americans and non-Hispanic Blacks were proportionally more represented among hypertensive participants, whereas non-Hispanic Whites were more prevalent in the non-hypertensive group (P<0.001).

Educational attainment differed between groups, with hypertension being more common among participants with lower education levels (primary or middle school) compared to those with a college degree or above (P<0.001). In terms of marital status, individuals who were married or divorced showed higher rates of hypertension than those who were never married (P<0.001).

The average BMI was markedly higher in the hypertensive group (31.1 ± 7.6 kg/m²) than in the non-hypertensive group (29.0 ± 6.9 kg/m², P<0.001). Likewise, mean systolic blood pressure was elevated in hypertensive individuals (132.6 ± 20.7 mmHg vs. 124.2 ± 18.8 mmHg, P<0.001), while diastolic pressure was slightly lower (71.4 ± 14.5 mmHg vs. 72.4 ± 13.1 mmHg, P=0.021).

Lifestyle behaviors also varied significantly: smoking was more prevalent among hypertensive participants (52.0% vs. 37.5%, P<0.001), while alcohol consumption showed no statistically significant difference (P=0.067). Average sodium intake was notably lower among hypertensive participants (3,207.3 ± 1,819.5 mg/day) compared with non-hypertensive individuals (3,560.2 ± 1,985.5 mg/day, P<0.001), whereas potassium intake did not differ significantly between groups.

Overall, hypertensive participants tended to be older, more likely female, have higher BMI, and exhibit distinct sociodemographic and behavioral profiles compared with non-hypertensive individuals (Table 1).

### Association between sodium and potassium intake and hypertension

In the multivariable logistic regression model adjusting for age, gender, race/ethnicity, education level, smoking status, alcohol consumption, and other relevant covariates, both sodium and potassium intake were not significantly associated with the odds of having hypertension. Specifically, the OR for sodium intake was 1.000 (95% CI: 1.000–1.000, P=0.820), and for potassium intake was 1.000 (95% CI: 1.000–1.000, P=0.957), suggesting no significant linear relationship between these dietary factors and hypertension in the overall population.

Among the covariates, older age (OR = 1.064, 95% CI: 1.059–1.069, P<0.001), male gender (OR = 1.893, 95% CI: 1.622–2.211, P<0.001), and certain racial/ethnic groups (e.g., Group 3: OR = 1.747, 95% CI: 1.348–2.274) were significantly associated with increased hypertension risk. Alcohol consumption was also associated with elevated odds (OR = 1.301, 95% CI: 1.008–1.687, P=0.045), whereas current smoking (SMQ020) was linked to a lower odds of hypertension (OR = 0.618, 95% CI: 0.530–0.719, P<0.001).

These findings are detailed in Table 2 and visually summarized in the forest plot (Fig 1), providing a clear overview of the associations between dietary sodium/potassium intake and hypertension, alongside other sociodemographic and lifestyle covariates. The near-unity odds ratios observed in these models likely reflect the use of standardized (z-scored) dietary variables, where one-unit changes correspond to one standard deviation in intake. This scaling improves comparability across nutrients but naturally yields values close to 1.00 when the underlying effects are modest. In addition, although the overall sample size provided sufficient power to detect moderate associations in the full population, the power for detecting weak effects or interactions in stratified analyses was limited. Therefore, these non-significant findings should be interpreted with caution rather than as definitive evidence of no association.

**Table 1. Baseline characteristics of participants by hypertension status.**

| Characteristics | hypertension | | P value |
|---|---|---|---|
| | No (3874) | Yes (1695) | |
| Age (mean±SD) | 46.51 (17.16) | 62.91 (13.50) | <0.001 |
| Gender (%) | | | <0.001 |
| Male | 1977 (51.0) | 725 (42.8) | |
| Female | 1897 (49.0) | 970 (57.2) | |
| Race (%) | | | <0.001 |
| Non-Hispanic White | 576 (14.9) | 159 (9.4) | |
| Non-Hispanic Black | 377 (9.7) | 140 (8.3) | |
| Mexican American | 1176 (30.4) | 759 (44.8) | |
| Other Hispanic | 889 (22.9) | 409 (24.1) | |
| Other Race – Including Multiracial | 856 (22.1) | 228 (13.5) | |
| Education Level (%) | | | <0.001 |
| No formal education/Primary education | 315 (8.1) | 164 (9.7) | |
| Some middle school | 438 (11.3) | 200 (11.8) | |
| High school graduate | 891 (23.0) | 434 (25.6) | |
| Some college/Associate degree | 1205 (31.1) | 573 (33.8) | |
| College graduate/Bachelor's degree | 1016 (26.2) | 320 (18.8) | |
| Master's degree | 1 (0.1) | 1 (0.1) | |
| Doctorate | 8 (0.2) | 3 (0.2) | |
| Marital Status (%) | | | <0.001 |
| Never married | 1891 (48.8) | 846 (49.8) | |
| Married | 191 (4.9) | 271 (16.0) | |
| Divorced | 380 (9.8) | 261 (15.4) | |
| Widowed | 135 (3.5) | 67 (4.0) | |
| Separated | 855 (22.1) | 151 (8.9) | |
| Other | 419 (10.8) | 96 (5.7) | |
| Refused | 3 (0.1) | 3 (0.2) | |
| BMI (mean±SD) | 28.99 (6.90) | 31.06 (7.59) | <0.001 |
| Systolic BP (mean±SD) | 124.22 (18.79) | 132.58 (20.73) | <0.001 |
| Diastolic BP (mean±SD) | 72.41 (13.12) | 71.41 (14.50) | 0.021 |
| Smoking Status (%) | | | <0.001 |
| Smoker | 1454 (37.5) | 881 (52.0) | |
| Non-smoker | 2420 (62.5) | 814 (48.0) | |
| Alcohol Consumption (%) | | | 0.067 |
| No | 371 (11.0) | 138 (9.2) | |
| Yes | 3011 (89.0) | 1364 (90.8) | |
| Physical Activity (%) | | | NA |
| No | 3874 (100.0) | 1695 (100.0) | |
| Sodium Intake (mean±SD) | 3560.16 (1985.45) | 3207.30 (1819.50) | <0.001 |
| Potassium Intake (mean±SD) | 2595.75 (1319.20) | 2494.61 (1239.98) | 0.013 |
| Total Energy Intake (mean±SD) | 2166.99 (1072.78) | 1984.25 (924.75) | <0.001 |
| Fiber Intake (mean±SD) | 17.31 (11.33) | 15.79 (10.16) | <0.001 |
| Cholesterol Intake (mean±SD) | 312.83 (260.52) | 294.46 (235.59) | 0.021 |

**Table 2. Multivariable logistic regression results for hypertension (Model 2).**

| Variables | OR (95%CI) | SE | Z value | P-value |
|---|---|---|---|---|
| Ref | 0.01(0.01-0.02) | 0.272 | −17.407 | <0.001 |
| Sodium Intake (mg/day) | 1.00(1.00-1.00) | 0.000 | 0.227 | 0.820 |
| Potassium Intake (mg/day) | 1.00(1.00-1.00) | 0.000 | 0.054 | 0.957 |
| Alcohol Consumption (Yes) | 1.30(1.09-1.69) | 0.131 | 2.009 | 0.045 |
| Smoking Status | 0.62(0.53-0.78) | 0.078 | −6.177 | <0.001 |
| Age (years) | 1.06(1.06-1.07) | 0.003 | 24.583 | <0.001 |
| Gender (Male) | 1.89(1.62-2.21) | 0.079 | 8.079 | <0.001 |
| Race/Ethnicity | | | | |
| Mexican American | 1.30(0.95-1.79) | 0.162 | 1.622 | 0.105 |
| Other Hispanic | 1.75(1.35-2.28) | 0.133 | 4.188 | <0.001 |
| Non-Hispanic White | 1.64(1.25-2.17) | 0.140 | 3.556 | <0.001 |
| Non-Hispanic Black | 1.28(0.95-1.73) | 0.152 | 1.633 | 0.102 |
| Education Level | | | | |
| 9–11th grade | 0.75(0.53-1.05) | 0.173 | −1.688 | 0.091 |
| High school graduate or GED equivalent | 0.89(0.66-1.28) | 0.155 | −0.695 | 0.487 |
| Some college or AA degree | 0.96(0.72-1.29) | 0.151 | −0.244 | 0.807 |
| College graduate or above | 0.61(0.45-0.83) | 0.160 | −3.102 | 0.002 |
| Refused to answer | 1.55(0.05-47.57) | 1.566 | 0.280 | 0.779 |
| Don't know | 0.19(0.01-1.34) | 1.134 | −1.461 | 0.144 |

Note: OR = Odds Ratio; CI = Confidence Interval; Ref = Reference group. Gender reference: male; Smoking: non-smoker; Race: Other Race/Multi-Racial; Education: Level 2 (Less than 9th grade) used as reference.

For interpretability, one standard deviation of sodium intake in this study corresponded to approximately 1,200 mg/day, while one standard deviation of potassium intake corresponded to approximately 600 mg/day. Analysees conducted on the raw intake scale yielded effect estimates in the same direction, but with very small magnitudes, consistent with the near-unity odds ratios observed after standardization.

## Sensitivity analysis

Across four progressively adjusted models, the estimated odds ratios for sodium and potassium remained near unity with narrow confidence intervals, indicating consistent lack of statistically significant associations (Fig 2).

## Expanded dietary intake variables

To further explore the associations between various dietary components and hypertension, we conducted a multivariable logistic regression including additional standardized dietary variables: total energy intake, fiber, and cholesterol, alongside sodium and potassium (Fig 3). In a multivariable model including standardized energy, fiber, cholesterol, sodium, and potassium, none of the dietary variables showed statistically significant associations with hypertension. Fiber demonstrated a modest inverse trend (95% CI marginally included the null), and the sodium-to-potassium (Na/K) ratio was positively but not significantly associated (OR = 1.10, 95% CI: 0.96–1.26; P = 0.18), consistent with the primary results.

These findings underscore that, within this population, individual dietary macronutrients- including fiber, cholesterol, and energy-were not independently associated with hypertension when controlling for multiple sociodemographic and behavioral confounders.Because several dietary variables (e.g., energy intake, sodium, potassium, fiber, and the sodium-to-potassium ratio) are interrelated, multicollinearity may be present in fully adjusted models. Such collinearity could attenuate individual effect estimates and contribute to the observed lack of statistically significant associations.

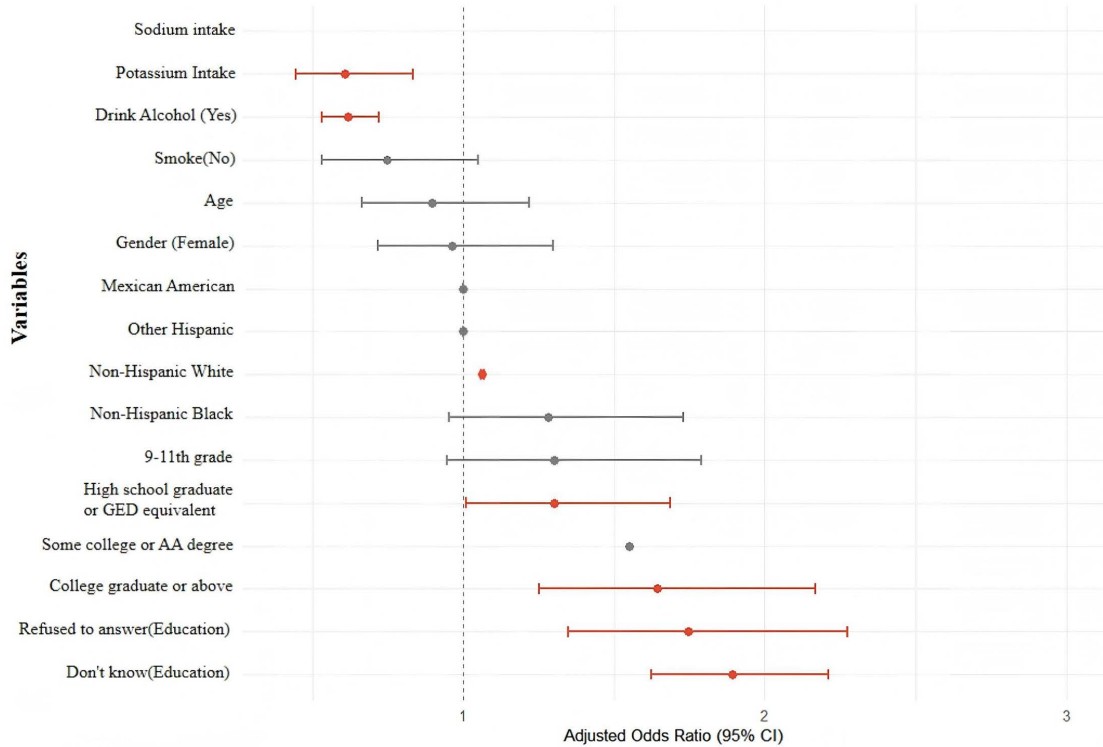

**Fig 1. Forest plot of multivariable-adjusted OR (95% CI) for hypertension associated with dietary intake and covariates.** The forest plot displays adjusted OR and 95% CI)for the association between sodium and potassium intake, demographic factors, education level, smoking, and alcohol consumption with hypertension. Significant associations ($P < 0.05$) are highlighted in red. Confidence intervals appear narrow because odds ratios are close to unity and the x-axis is scaled accordingly.

## Dose–response relationship via RCS analysis

To examine potential nonlinear associations between continuous dietary intake variables and hypertension, we employed RCS regression models for sodium and potassium intake (Fig 4).

For sodium intake (Fig 4A), the RCS plot showed a relatively flat curve at lower intake levels, followed by a modest upward trend in odds ratios at higher sodium intake levels. However, the 95% confidence interval bands were wide, indicating substantial variability and a lack of statistically robust nonlinear association. No apparent threshold effect was observed across the full intake distribution.

For potassium intake (Fig 4B), the RCS analysis revealed a nearly flat or slightly downward curve, suggesting a weak inverse association with hypertension risk. Similar to sodium, confidence intervals remained wide across intake ranges, indicating limited statistical power to detect nonlinear effects. The curve did not reveal a clear J- or U-shaped pattern, and the overall trend was largely stable.

These results suggest no strong evidence of nonlinear associations between sodium or potassium intake and hypertension within this population.

## Subgroup analyses

To explore whether the association between dietary intake and hypertension varied across population subgroups, stratified analyses were conducted by gender, age, and BMI categories (Fig 5A–5C).

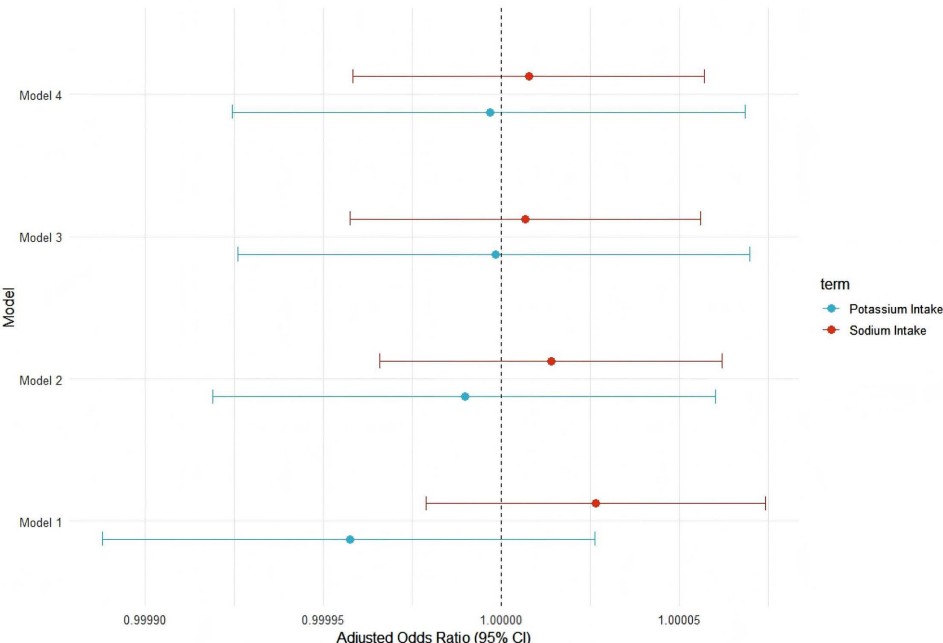

**Fig 2. Sensitivity analysis of adjusted OR (95% CI) for hypertension by sodium and potassium intake across four multivariate models.** Multivariable logistic regression models (Model 1 to Model 4) were used with progressively adjusted covariates. Model 1: unadjusted; Model 2: adjusted for age and gender; Model 3: additionally adjusted for race/ethnicity, education level, and marital status; Model 4: further adjusted for smoking status, alcohol consumption, and physical activity.

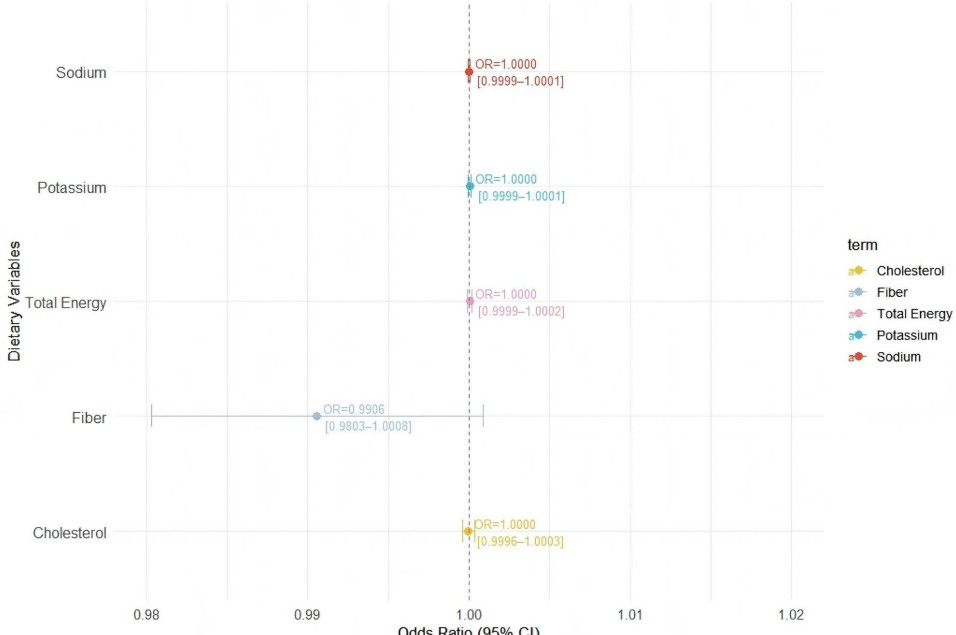

**Fig 3. Adjusted OR (95% CI) for hypertension associated with standardized dietary intake variables, including sodium, potassium, fiber, cholesterol, and total energy.**

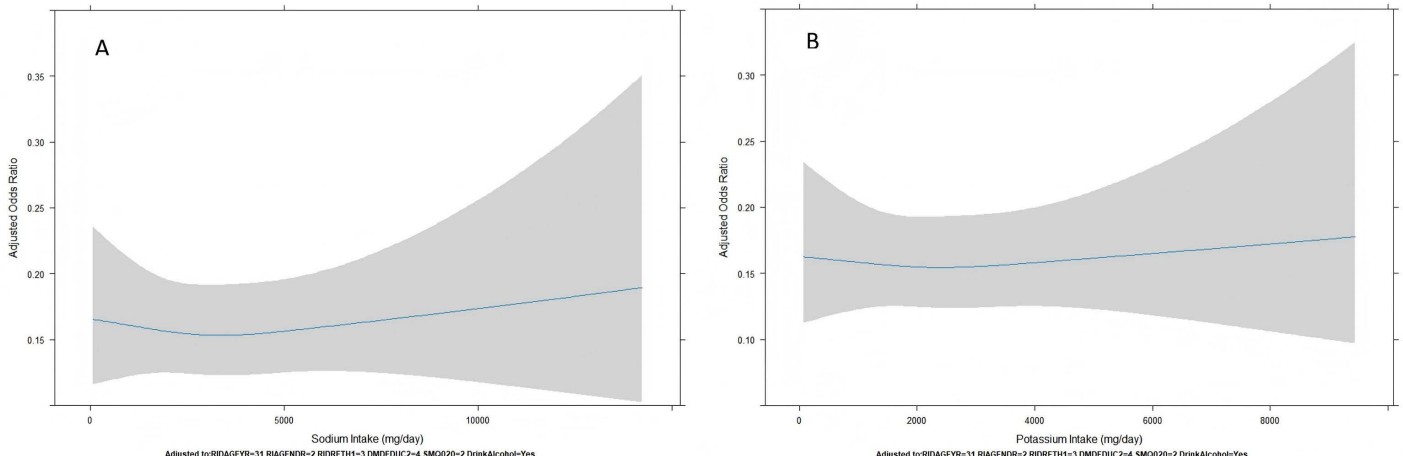

**Fig 4. Restricted cubic spline plot of the association between sodium (A) and potassium (B) intake and hypertension.** Restricted cubic spline models were fitted using the median intake level as the reference value (OR = 1.0). The x-axis represents dietary intake in mg/day.

In gender-stratified analyses, sodium and potassium intake showed no statistically significant associations with hypertension in either males or females. Confidence intervals were wide and largely overlapping across groups, providing no evidence of a statistically significant gender-based interaction. Similar patterns were observed for fiber and total energy intake (Fig 5A). When stratified by age (<60 vs. ≥ 60 years), no statistically significant associations between sodium or potassium intake and hypertension were observed in either age group. Although point estimates varied slightly across age strata for some dietary variables, confidence intervals overlapped substantially, and no statistically significant interactions were detected (Fig 5B). In BMI-stratified analyses (normal weight vs. overweight/obese), point estimates for sodium, fiber, and cholesterol differed slightly between groups; however, confidence intervals were wide and overlapping, and none of the interaction terms reached statistical significance (all P for interaction > 0.05), indicating limited evidence of effect modification by BMI status (Fig 5C).

Overall, these stratified analyses indicate that the associations between dietary factors and hypertension were broadly consistent across gender, age, and BMI categories, with no evidence of statistically significant effect modification.

## Diet–blood pressure relationship by hypertension status

To explore whether associations between dietary intake and blood pressure differed by hypertension status, analyses were stratified by self-reported history of hypertension. As shown in Fig 6A, sodium intake was positively associated with DBP in both hypertensive and non-hypertensive participants, while potassium intake showed an inverse association with DBP in both groups. However, confidence intervals were wide and largely overlapping, and no statistically significant interaction by hypertension status was observed.

Similar patterns were observed for SBP (Fig 6B). Higher sodium intake was associated with higher SBP, whereas potassium intake was inversely associated with SBP in both groups. Although point estimates differed slightly between participants with and without hypertension, confidence intervals overlapped substantially, indicating considerable uncertainty around subgroup-specific estimates.

Overall, these stratified analyses provide no evidence of statistically significant effect modification by hypertension status. Observed differences in point estimates should therefore be interpreted as descriptive rather than indicative of meaningful subgroup differences.

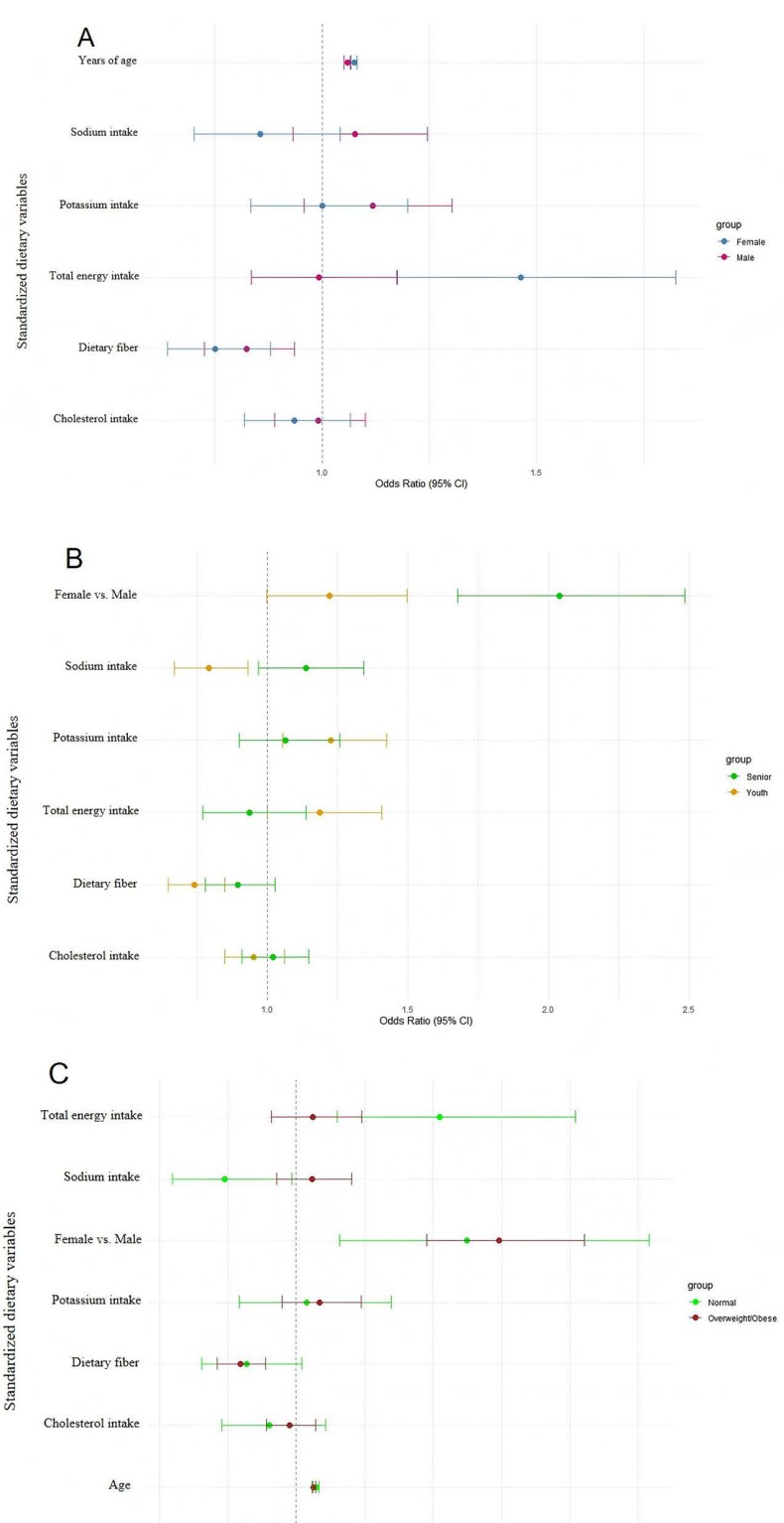

**Fig 5. A. Subgroup analysis of dietary intake and hypertension stratified by gender. B.** Subgroup analysis of dietary intake and hypertension stratified by age group (<60 vs. ≥60). **C.** Subgroup analysis of dietary intake and hypertension stratified by BMI status. No statistically significant interactions were detected across subgroups (all P for interaction > 0.05).

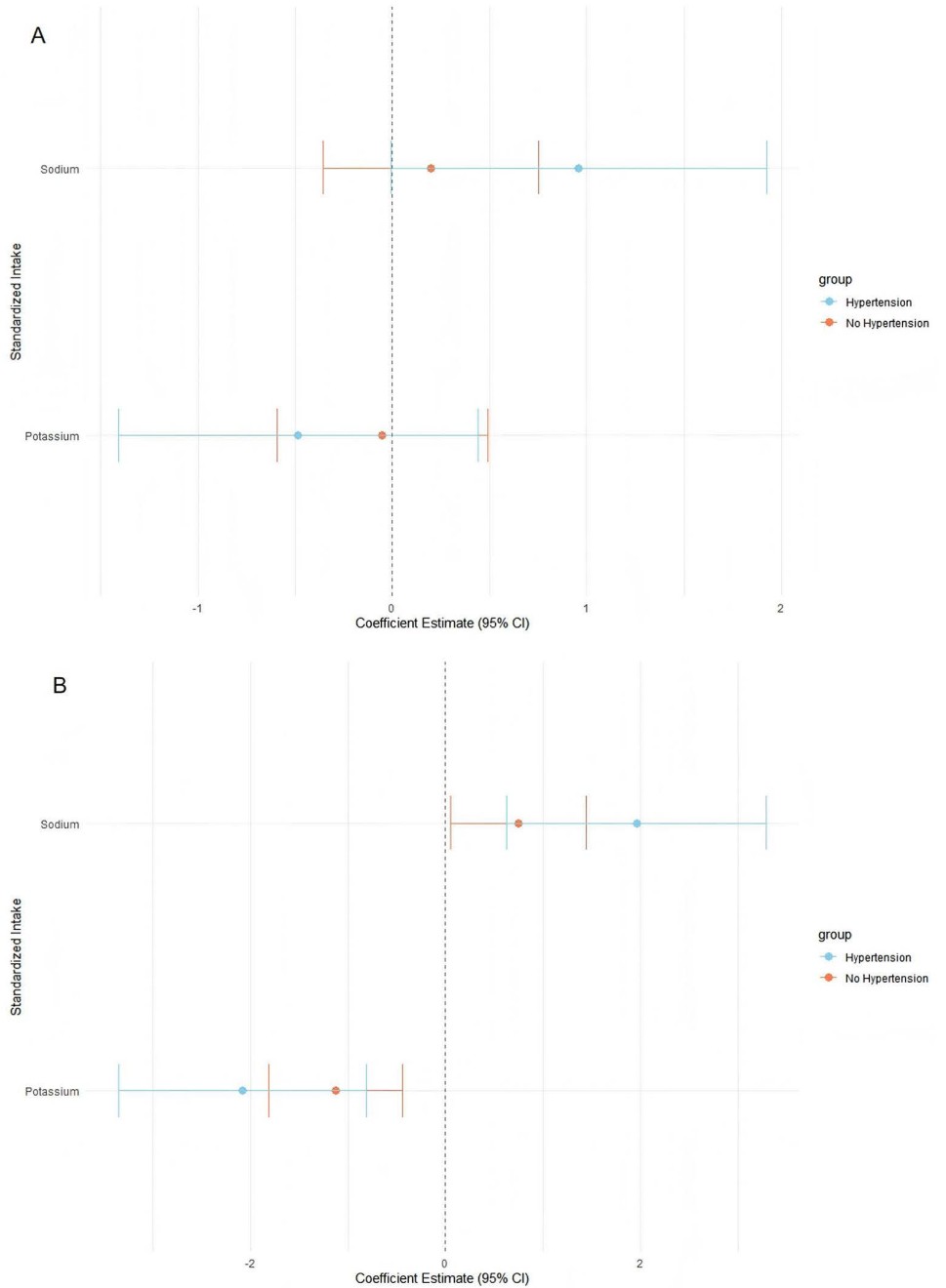

**Fig 6. Diet-blood pressure relationship by hypertension status.**

## Discussion

In this nationally representative cross-sectional study based on NHANES 2017–2018 data, we investigated the associations between dietary sodium and potassium intake and hypertension risk. Contrary to our initial hypothesis, neither sodium nor potassium intake showed statistically significant associations with the hypertension prevalence after adjusting

for key sociodemographic and behavioral covariates. Furthermore, when the sodium-to-potassium (Na/K) ratio was incorporated into the model, the results remained unchanged, showing a positive but non-significant association with hypertension. This consistency indicates that the null findings are not attributable to the omission of this composite exposure metric. These findings remained robust across multiple sensitivity models with progressive covariate adjustments.

While sodium and potassium intake did not demonstrate significant independent effects, a suggestive inverse association was observed between dietary fiber intake and hypertension, although the confidence interval marginally included the null value. Moreover, subgroup analyses and restricted cubic spline modeling failed to reveal strong effect modification or nonlinear trends, respectively. Taken together, these findings indicated that within typical dietary intake ranges observed in the U.S. population, sodium and potassium might lack independent predictive utility for hypertension after adjustment for key lifestyle and demographic confounders. These results emphasize the need to contextualize dietary guidelines within broader behavioral and metabolic patterns. The modest inverse association observed for dietary fiber aligns with prior evidence linking higher fiber consumption to lower blood pressure. Mechanistically, dietary fiber could improve endothelial function, enhance sodium excretion and modulate gut microbiota–derived metabolites such as short-chain fatty acids, thereby supporting vascular homeostasis and attenuating systemic inflammation. This protective association is corroborated by numerous observational studies (including analyses of NHANES data) and randomized controlled trials investigating the effects of fiber supplementation on hypertension risk [19–21].

Numerous epidemiological and interventional studies have demonstrated a positive association between sodium intake and blood pressure—evidence that underpins global public health recommendations advocating sodium reduction to prevent hypertension [9,22,23]. For instance, the INTERSALT study and data synthesized by the World Health Organization have support a linear dose–response relationship between sodium intake and blood pressure elevation [15]. However, our findings did not replicate this association, potentially reflecting differences in study design, population characteristics, and methods of dietary assessment.

Regarding potassium, prior literature has consistently suggested a protective effect against hypertension by promoting natriuresis, reducing vascular resistance and improving endothelial function mechanistically [24]. We observed an inverse trend but without statistical significance, possibly due to limited power or residual confounding. Similarly, our exploratory analysis indicated a potential protective role of dietary fiber—an effect on insulin sensitivity, inflammation and vascular health that has been supported by several cohort studies [25]. These discrepancies highlight the complexity of diet-disease relationships and underscore the importance of considering total dietary patterns, individual variability and measurement precision in nutritional epidemiology.

Although our primary analyses did not reveal significant associations between sodium or potassium intake and hypertension, further stratified and nonlinear modeling provided additional insights. In subgroup analyses stratified by gender, age, and BMI, no statistically significant interaction effect was observed. However, we noted that certain subpopulations—particularly older adults and individuals with higher BMI—exhibited numerically stronger associations between dietary intake and hypertension risk. These findings suggest the possibility of differential susceptibility to dietary factors among high-risk groups, warranting further targeted research.

The RCS models did not support a nonlinear relationship between sodium or potassium intake and hypertension. The risk curves were relatively flat across intake ranges, with wide confidence intervals and no evident threshold or J/U-shaped patterns. These findings challenge the assumption of a clear dose-response or threshold effect and imply that, within the commonly observed intake range in the U.S. population, variations in sodium and potassium intake may have limited predictive value for hypertension risk in isolation.

Our findings contribute to the growing body of literature questioning the universality of sodium and potassium intake thresholds for hypertension risk reduction. Several factors may help explain the absence of statistically significant associations observed in this study. Firstly, dietary sodium and potassium intake were estimated from a single 24-hour recall, which was prone to random measurement error and might not accurately represent habitual intake. Such nondifferential

misclassification would tend to bias estimates toward the null. Secondly, although extensive covariate adjustment was performed, residual confounding from unmeasured lifestyle or metabolic factors could not be completely excluded. Finally, the variability in sodium and potassium intake within the population was relatively limited, potentially reducing the statistical power to detect modest associations. These considerations may partly account for the observed lack of significant findings.

This study has several notable strengths. Firstly, it utilizes data from the NHANES 2017–2018 cycle, a nationally representative and rigorously collected dataset with a complex sampling design, enhancing the generalizability of our findings to the U.S. adult population. Secondly, we employed a comprehensive analytical strategy, including multivariable logistic regression, sensitivity analyses with progressively adjusted models, RCS modeling and stratified subgroup analyses. These approaches ensured robust estimation and allowed for exploration of potential nonlinearities and population-specific patterns. Thirdly, the inclusion of multiple dietary variables beyond sodium and potassium, such as total energy, fiber, and cholesterol intake, provided a more nuanced assessment of diet–blood pressure relationships.

However, several limitations should be acknowledged. The cross-sectional design precludes any causal inference regarding the relationship between dietary intake and hypertension. Reverse causation is also possible, as individuals diagnosed with hypertension may have altered their dietary behaviors in response to medical advice. These behavioral changes could attenuate the observed associations between nutrient intake and hypertension. Moreover, biomarkers such as urinary sodium or potassium excretion were not included, limiting the precision of intake estimation. In addition, residual confounding cannot be ruled out despite multivariable adjustments, particularly for unmeasured variables such as sodium-to-potassium ratio, dietary patterns, or genetic predisposition. Finally, because self-reported hypertension may underestimate true prevalence, particularly among younger individuals or those with limited access to health care, this form of outcome misclassification is likely to be nondifferential and to bias observed associations toward the null.

While sodium reduction and potassium enhancement remain cornerstone recommendations in public health nutrition, our results imply that the effecacy vary depending on context, such as baseline dietary habits, stage of blood pressure elevation, and individual susceptibility. Although stratified or sensitivity analyses excluding known hypertensive participants could partly address this concern, such approaches would markedly reduce the sample size and population representativeness. Future longitudinal and interventional studies are warranted to confirm the temporal relationship between dietary electrolyte intake and hypertension development.

## Conclusion

In this cross-sectional analysis of a nationally representative sample from NHANES 2017–2018, we found no significant associations between dietary sodium or potassium intake and hypertension after adjusting for major demographic and behavioral covariates. These null findings remained consistent across multiple sensitivity models, subgroup analyses, and nonlinear dose–response evaluations. Although exploratory analyses suggested a potential inverse trend between dietary fiber and hypertension, the evidence was not statistically conclusive.

Our results underscore the complexity of diet–blood pressure relationships and suggest that sodium and potassium, when assessed individually and within typical intake ranges, may not serve as independent predictors of hypertension in the general population. These findings highlight the need for more comprehensive dietary evaluations that consider nutrient interactions, habitual dietary patterns, and individual variability. Future longitudinal studies incorporating objective biomarkers and precision nutrition approaches are warranted to better inform dietary guidelines and public health strategies for hypertension prevention. Given the cross-sectional nature of this study, causal inferences cannot be established, and longitudinal investigations are required to confirm these associations.

## Author contributions

**Conceptualization:** Xiao Luo.

**Data curation:** Longfeng Ran, Zhuojun Kang.

**Formal analysis:** Longfeng Ran, Zhuojun Kang.

**Funding acquisition:** Xin Mu.

**Methodology:** Xiao Luo.

**Project administration:** Xin Mu.

**Software:** Xuan Zhang, Xinyu Fu.

**Supervision:** Xin Mu.

**Validation:** Xuan Zhang, Xinyu Fu.

**Writing – original draft:** Xiao Luo.

**Writing – review & editing:** Xin Mu.

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
