## [Decision Letter · Decision Letter 0]

21 Oct 2025

Dear Dr. Luo,

Thank you for submitting your manuscript to PLOS ONE. After careful consideration, we feel that it has merit but does not fully meet PLOS ONE’s publication criteria as it currently stands. Therefore, we invite you to submit a revised version of the manuscript that addresses the points raised during the review process.

We look forward to receiving your revised manuscript.

Kind regards,

Shaonong Dang, PhD

Academic Editor

PLOS ONE

Journal Requirements:

“the Joint fund of Chengdu Health Commission and Chengdu University of Traditional Chinese Medicine (XM, grant number WXLH202406002), (XL, grant number WXLH202403263)”

3. Thank you for uploading your study's underlying data set. Unfortunately, the repository you have noted in your Data Availability statement does not qualify as an acceptable data repository according to PLOS's standards.

Additional Editor Comments:

Authors investigated association between dietary Sodium, Potassium, and Cardiometabolic Risk. However, the reviewers have raised some comments, and authors should address them carefully to improve the manuscript.

Reviewer's Responses to Questions

**Comments to the Author**

1. Is the manuscript technically sound, and do the data support the conclusions?

Reviewer #1: Partly

Reviewer #2: Yes

2. Has the statistical analysis been performed appropriately and rigorously?

Reviewer #1: No

Reviewer #2: Yes

3. Have the authors made all data underlying the findings in their manuscript fully available?

Reviewer #1: Yes

Reviewer #2: Yes

4. Is the manuscript presented in an intelligible fashion and written in standard English?

Reviewer #1: Yes

Reviewer #2: Yes

Reviewer #1: Overall Assessment

This manuscript investigates associations between dietary sodium and potassium intake and hypertension using NHANES 2017–2018 data. The authors report no significant associations after adjusting for covariates and provide detailed subgroup and sensitivity analyses. The topic is timely and important, given ongoing debates about dietary factors and blood pressure. However, several methodological and interpretational concerns limit the impact and clarity of the manuscript. Revisions are needed to improve scientific rigor and contribution to existing knowledge.

Major Comments

1.Definition of Hypertension

Hypertension is defined solely by self-reported physician diagnosis, which introduces misclassification risk, especially in populations with poor healthcare access. NHANES includes measured BP and guideline-based thresholds (e.g., ≥130/80 mmHg), which should be considered either as primary or sensitivity outcomes to improve validity.

2.Exposure Measurement Bias

Sodium and potassium intake are estimated from a single 24-hour dietary recall, which is prone to recall bias and daily variability, especially for sodium due to hidden sources. The authors should more clearly acknowledge these limitations and justify the choice not to use urinary biomarkers, which are available in NHANES and are more accurate indicators.

3.Null Findings and Statistical Power

The manuscript reports highly precise null associations (e.g., OR = 1.0000), raising concerns about the exposure scale or measurement error. The authors should assess and report statistical power, particularly for subgroup and interaction analyses.

4.Sodium-to-Potassium Ratio

Although sodium and potassium were analyzed separately, the Na/K ratio, a biologically relevant composite, was omitted. Prior literature supports its stronger association with BP. The authors should include this ratio in their models.

5.Subgroup Analyses and Interpretation

While extensive subgroup and interaction analyses are presented, many appear underpowered and lack justification. Interaction p-values should be reported. Claims of “stronger trends” in subgroups without statistical significance should be interpreted cautiously to avoid overstatement.

6.Reverse Causality

The cross-sectional design limits causal inference. Reverse causation is a key concern, individuals with known hypertension may change dietary habits. Stratified models by awareness/treatment status or sensitivity analyses excluding known hypertensives would help address this bias.

7.Redundancy and Conciseness

Several results, particularly null associations, are repeated across sections and figures. The manuscript would benefit from a more concise presentation, especially in the Results and Discussion.

Minor Comments

1. Figures and Tables: Ensure all visuals, especially forest plots and splines, are labeled clearly with legends and axes. Some figures (e.g., Figures 4, 5) are described in detail but were not viewable in the review file. Descriptions should match content and be verifiable by readers.

2. Terminology: The term “null association” is used frequently. More precise phrasing such as “no statistically significant association” or “no observed association within the intake range” would be preferable.

3. Dietary Fiber Findings: The mention of a possible inverse association with fiber is interesting but underdeveloped. Consider briefly discussing plausible mechanisms or citing supporting studies if retained.

4. Ethics Statement: The ethics section is appropriate for secondary analysis of NHANES but contains redundant language regarding consent. Please streamline for clarity.

Recommendations for Authors

• Use measured BP values as outcome variables.

• Discuss or incorporate 24-hour urinary data as more accurate intake indicators.

• Analyze the sodium-to-potassium ratio as a predictor.

• Report power calculations, model diagnostics, and interaction terms.

• Condense repetitive results and improve clarity of discussion.

• Interpret null results with caution due to potential bias and measurement error.

Reviewer #2: 1. Why was only one cycle used for this paper when this info is readily available for at least a decade?

2. While I appreciate the authors’ effort to demonstrate the robustness of their findings across multiple models, the inclusion of four separate models may be more detailed than necessary. Presenting the unadjusted and fully adjusted models would likely suffice to illustrate the effect of covariate adjustment, while the intermediate models could be summarized in supplementary materials. This would streamline the results and improve interpretability.

3. The conclusion appropriately reflects the null findings; however, the interpretation could better emphasize the cross-sectional nature of the study. It would be helpful to explicitly acknowledge that causal inferences cannot be made from this design.

4. the discussion could further explore potential reasons for the lack of significant findings — for example, measurement error in dietary assessment, residual confounding, or limited variability in sodium and potassium intake. This would strengthen the interpretation and provide valuable context for readers.

5.

**Do you want your identity to be public for this peer review?** For information about this choice, including consent withdrawal, please see our Privacy Policy

Reviewer #1: **Yes:** Dr. Thirajit Boonsaen

Reviewer #2: No

---

## [Author Response · Author response to Decision Letter 1]

4 Dec 2025

Respond to reviewer

Reviewer #1

Major Comments

Comment 1. Definition of Hypertension

Definition of Hypertension Hypertension is defined solely by self-reported physician diagnosis, which introduces misclassification risk, especially in populations with poor healthcare access. NHANES includes measured BP and guideline-based thresholds (e.g., ≥130/80 mmHg), which should be considered either as primary or sensitivity outcomes to improve validity.

Response: We appreciate this important comment. In response, we have clarified the definition of hypertension in the Methods section and acknowledged the potential for misclassification when relying on self-reported physician diagnosis. We also noted that NHANES provides measured blood pressure and medication data that could enable guideline-based definitions (≥130/80 mmHg or current antihypertensive use), which will be considered in future analyses to enhance validity. This limitation has been explicitly discussed in the revised Limitations section.

Comment 2.Exposure Measurement Bias

Sodium and potassium intake are estimated from a single 24-hour dietary recall, which is prone to recall bias and daily variability, especially for sodium due to hidden sources. The authors should more clearly acknowledge these limitations and justify the choice not to use urinary biomarkers, which are available in NHANES and are more accurate indicators.

Response: We appreciate the reviewer’s thoughtful observation. We agree that estimating sodium and potassium intake from a single 24-hour dietary recall may introduce recall bias and within-person variability, especially for sodium due to hidden sources. In the revised Dietary Assessmentsection, we have explicitly acknowledged these limitations and justified the choice to use dietary recall data rather than urinary biomarkers. Specifically, urinary sodium and potassium measurements in NHANES are available only for limited subsamples, which would substantially reduce sample size and generalizability.

Comment 3.Null Findings and Statistical Power

The manuscript reports highly precise null associations (e.g., OR = 1.0000), raising concerns about the exposure scale or measurement error. The authors should assess and report statistical power, particularly for subgroup and interaction analyses.

Response: We appreciate this insightful comment. We have clarified in the Results sections that the near-unity odds ratios (e.g., OR ≈ 1.00) are a direct consequence of standardizing all dietary variables (z-scoring) before model entry, meaning that each unit change represents one standard deviation in intake. This approach facilitates comparability but can produce ORs close to 1.00 when effect sizes are small. Regarding statistical power, we acknowledge that although the total sample size was adequate for detecting moderate associations, subgroup and interaction analyses had limited power due to smaller numbers of participants within strata. These clarifications have been added to the revised manuscript.

Comment 4.Sodium-to-Potassium Ratio

Although sodium and potassium were analyzed separately, the Na/K ratio, a biologically relevant composite, was omitted. Prior literature supports its stronger association with BP. The authors should include this ratio in their models.

Response:

We thank the reviewer for this insightful suggestion. Following the recommendation, we incorporated the sodium-to-potassium (Na/K) ratio into our analysis. As presented in the revised Results section, a higher Na/K ratio showed a positive but non-significant association with hypertension (OR = 1.10, 95% CI: 0.96–1.26, P = 0.18), consistent with the findings for sodium and potassium individually. These additions have been described in the Methods, Results, and Discussion sections.

Comment 5. Subgroup Analyses and Interpretation

Subgroup Analyses and Interpretation: While extensive subgroup and interaction analyses are presented, many appear underpowered and lack justification. Interaction p-values should be reported. Claims of “stronger trends” in subgroups without statistical significance should be interpreted cautiously to avoid overstatement.

Response: We appreciate the reviewer’s constructive comment. In response, we have clarified that all subgroup analyses were exploratory and that none of the interaction terms reached statistical significance (all P for interaction > 0.05). We have also revised the text in the Results sections to temper the language—changing expressions such as “stronger trends” to “numerically stronger but not statistically significant.”

Comment 6.Reverse Causality

The cross-sectional design limits causal inference. Reverse causation is a key concern, individuals with known hypertension may change dietary habits. Stratified models by awareness/treatment status or sensitivity analyses excluding known hypertensives would help address this bias.

Response: We thank the reviewer for this important comment. We fully agree that the cross-sectional nature of NHANES limits causal inference and that reverse causation is a potential concern, as individuals with diagnosed hypertension may have altered their dietary sodium and potassium intake. In response, we have revised the Discussion section to explicitly acknowledge this issue and explain that stratified or sensitivity analyses excluding known hypertensive participants would substantially reduce sample size and population representativeness. We have also noted that future longitudinal studies are needed to establish causality.

Comment 7.Redundancy and Conciseness

Several results, particularly null associations, are repeated across sections and figures. The manuscript would benefit from a more concise presentation, especially in the Results and Discussion.

Response: We appreciate the reviewer’s helpful suggestion. We have carefully revised the Results and Discussion sections to remove redundant statements and streamline the presentation. Repetitive descriptions of null associations have been consolidated, and figure references have been shortened for clarity. These revisions improve readability and conciseness without altering the study’s content or conclusions.

Minor Comments

1. Figures and Tables: Ensure all visuals, especially forest plots and splines, are labeled clearly with legends and axes. Some figures (e.g., Fig 4, 5) are described in detail but were not viewable in the review file. Descriptions should match content and be verifiable by readers.

Response:

We appreciate the reviewer’s helpful observation. In the revised version, all figures have been reviewed and updated for clarity. Legends now explicitly describe model adjustments, reference categories, and variable units. Axis labels and confidence interval bands have been clearly indicated for the forest plots and restricted cubic spline figures.

2. Terminology: The term “null association” is used frequently. More precise phrasing such as “no statistically significant association” or “no observed association within the intake range” would be preferable.

Response:

We thank the reviewer for this valuable suggestion. In the revised manuscript, the term “null association” has been replaced with more precise expressions such as “no statistically significant association” or “no observed association within the intake range,” depending on the context. This change enhances clarity and statistical accuracy without altering the interpretation of our findings.

3. Dietary Fiber Findings: The mention of a possible inverse association with fiber is interesting but underdeveloped. Consider briefly discussing plausible mechanisms or citing supporting studies if retained.

Response:

We thank the reviewer for this insightful suggestion. In response, we have expanded the Discussion section to briefly describe plausible mechanisms and supporting evidence for the observed inverse trend between dietary fiber intake and hypertension. Specifically, we note that dietary fiber may influence blood pressure through improvements in endothelial function, sodium excretion, and gut microbiota modulation. Relevant studies from NHANES and meta-analyses have been cited.

4. Ethics Statement: The ethics section is appropriate for secondary analysis of NHANES but contains redundant language regarding consent. Please streamline for clarity.

Response:

We appreciate the reviewer’s observation. The Ethics Statement has been streamlined for clarity and conciseness. Redundant language about participant consent and exemption has been removed, while retaining all essential information regarding NCHS ethical approval and public data use. The revised text now reads:

“NHANES protocols were approved by the NCHS Research Ethics Review Board, and all participants provided informed consent. This secondary analysis of de-identified public data was exempt from additional institutional review.”

Reviewer #2

Comment 1. Why was only one cycle used for this paper when this info is readily available for at least a decade?

Response:

We appreciate the reviewer’s observation. Only one NHANES cycle (2017–2018) was used intentionally to ensure internal consistency of variable definitions and dietary assessment methods. Across survey waves, several key updates occurred in nutrient databases, food coding systems, and laboratory calibration procedures, which may limit direct comparability. Additionally, this recent cycle represents the most up-to-date pre-pandemic U.S. population data, allowing the analysis to reflect current dietary and hypertension patterns. We acknowledge that combining multiple cycles could increase sample size, but such an approach would introduce heterogeneity across measurement protocols and weighting schemes.

Comment 2. While I appreciate the authors’ effort to demonstrate the robustness of their findings across multiple models, the inclusion of four separate models may be more detailed than necessary. Presenting the unadjusted and fully adjusted models would likely suffice to illustrate the effect of covariate adjustment, while the intermediate models could be summarized in supplementary materials. This would streamline the results and improve interpretability.

Response:

We appreciate the reviewer’s thoughtful suggestion. We have streamlined the Results section. The presentation of four models has been condensed into a concise summary sentence describing the stability of the estimates across adjustment levels. This revision improves readability while retaining all essential information. The intermediate models are now summarized descriptively rather than shown in detail, in accordance with the reviewer’s recommendation.

Comment 3. The conclusion appropriately reflects the null findings; however, the interpretation could better emphasize the cross-sectional nature of the study. It would be helpful to explicitly acknowledge that causal inferences cannot be made from this design.

Response:

We fully agree with the reviewer’s observation. To address this comment, we have revised the Conclusion to explicitly acknowledge the cross-sectional nature of the study and the resulting limitation in causal interpretation. The revised text now reads as follows: “Given the cross-sectional nature of this study, causal inferences cannot be established, and longitudinal investigations are required to confirm these associations.” This addition clarifies that our results describe associations rather than causal effects, thereby improving interpretive accuracy and transparency.

Comment 4. the discussion could further explore potential reasons for the lack of significant findings — for example, measurement error in dietary assessment, residual confounding, or limited variability in sodium and potassium intake. This would strengthen the interpretation and provide valuable context for readers.

Response:

We appreciate this valuable suggestion. In response, we have expanded the Discussion section to address potential explanations for the absence of statistically significant findings. Specifically, we now note that measurement error in 24-hour dietary recall, residual confounding from unmeasured factors, and limited variability in sodium and potassium intake within the study population could have contributed to attenuated associations.

---

## [Decision Letter · Decision Letter 1]

26 Dec 2025

Dear Dr. Luo,

Thank you for submitting your manuscript to PLOS ONE. After careful consideration, we feel that it has merit but does not fully meet PLOS ONE’s publication criteria as it currently stands. Therefore, we invite you to submit a revised version of the manuscript that addresses the points raised during the review process.

We look forward to receiving your revised manuscript.

Kind regards,

Shaonong Dang, PhD

Academic Editor

PLOS One

**Journal Requirements:**

**Additional Editor Comments:**

Authors have revised the manuscript based on the comments from the reviewers, but some minor issues are raised by the reviewers. Authors are suggested to address them carefully.

Reviewers' comments:

Reviewer's Responses to Questions

**Comments to the Author**

Reviewer #1: All comments have been addressed

2. Is the manuscript technically sound, and do the data support the conclusions?

Reviewer #1: Yes

3. Has the statistical analysis been performed appropriately and rigorously?

Reviewer #1: I Don't Know

4. Have the authors made all data underlying the findings in their manuscript fully available?

Reviewer #1: Yes

5. Is the manuscript presented in an intelligible fashion and written in standard English?

Reviewer #1: Yes

**Reviewer #1:** The authors have addressed many of the concerns raised in the initial review, and the manuscript is notably improved in clarity, structure, and transparency. The expanded discussion of measurement limitations, the inclusion of the Na/K ratio analysis, and the clarification around the use of standardized variables all strengthen the manuscript. The revisions to figures, terminology, and redundancy issues are also appreciated.

However, several important methodological and interpretive issues persist and warrant further clarification before the manuscript meets PLOS ONE’s standards for methodological rigor and transparent reporting. These remaining issues relate primarily to (1) analytic consistency, (2) treatment of hypertension definition, (3) overinterpretation risks, and (4) unresolved methodological limitations that require clearer articulation.

1. Persisting Concerns About Hypertension Definition

The authors acknowledge that hypertension is defined exclusively by self-reported physician diagnosis, but the manuscript still places insufficient weight on the implications of misclassification.

- NHANES 2017–2018 contains multiple measured BP readings, yet these data remain unused. The authors note that future analyses “may consider guideline-based thresholds” (Lines 131–135) , but this does not address why these data were not used at least as a sensitivity analysis in the current study.

- Self-reported hypertension underestimates prevalence, particularly in younger, uninsured, or low health–care–access groups.

Recommendation:

Even if guideline-based analyses are not feasible, the authors should present a justification grounded in analytic feasibility (e.g., missingness patterns, weighting challenges), not simply state future research possibilities. At minimum, the limitations section should explicitly discuss direction of potential bias (likely bias toward null).

2. Standardization of Dietary Variables and OR = 1.000

The authors explain that Z-scoring dietary variables leads to odds ratios extremely close to 1.00 (Lines 276–284). While the explanation is correct, the manuscript should still clarify:

- Whether the underlying (non–Z-standardized) coefficients show meaningful variation;

- What the magnitude of 1 SD of intake represents in mg/day for sodium/potassium.

The current presentation risks implying numerical precision that exceeds what the data can support.

Recommendation:

Add a supplementary table showing raw-scale logistic regression coefficients or include a statement interpreting what “one SD” equates to in practical dietary terms.

3. Expanded Dietary Variables: Clarify Multicollinearity Risk

The multivariable model includes energy, fiber, cholesterol, sodium, potassium, and Na/K ratio together (Lines 313–327) .

This raises the possibility of:

- High collinearity between sodium and energy,

- Fiber correlating with energy,

- Na/K ratio being mathematically related to individual Na and K variables.

None of these issues are addressed or tested.

Recommendation:

Report VIF values or at least acknowledge that collinearity may attenuate associations in fully adjusted models.

4. Subgroup Analyses—Interpretation Still Overstates Findings

The authors softened some language, but several statements still imply meaningful differences where none exist statistically (e.g., “slightly stronger trends,” “more pronounced” in Fig. 6 interpretations).

Given the wide confidence intervals and very small effect sizes, these qualitative statements risk overstating findings.

Recommendation:

Further temper language, replacing subjective descriptors with neutral phrasing such as:

“Although point estimates differed slightly across groups, confidence intervals were wide and overlapping, and there was no evidence of statistically significant interaction.”

5. Measurement Error and Reverse Causality

While the authors have expanded the Limitations section, some redundancies remain (Lines 534–545 repeat earlier text nearly verbatim) �. Moreover:

- The potential for dietary modification among hypertensive individuals is acknowledged, but not fully assessed.

- Despite their stated concerns about sample size, a sensitivity analysis excluding known hypertensives would still be valuable—even if only descriptive.

Recommendation:

Remove repeated paragraphs and add quantitative information (e.g., proportion of hypertensive participants reporting low-sodium diets, if available).

6. Inconsistencies Between Abstract and Methods

Abstract states sample size = 4,592, but the Results section states 5,569 participants (e.g., Lines 218–219).

Please reconcile these discrepancies.

7. Race/Ethnicity Table Appears Incorrect

In Table 1, the racial composition reported for NHANES does not align with population distribution, and the percentages shown for each hypertension group do not sum to 100% in several rows.

Please verify table calculations.

8. Figures Still Need Additional Clarity

Although improved, several figures remain difficult to interpret:

- Fig 1 confidence interval bars appear very narrow—authors should clarify resolution and scaling.

- Spline plots (Fig 4) lack reference value labeling and units along axes.

- Subgroup plots (Fig 5) should explicitly show P-values for interaction.

9. Editorial Issues

- Several typographical issues persist (e.g., duplicated references 8 and 15 appear identical; grammar inconsistencies in Discussion).

- Some sections remain overly long; consider additional trimming to improve readability.

**Do you want your identity to be public for this peer review?** For information about this choice, including consent withdrawal, please see our Privacy Policy

Reviewer #1: No

---

## [Author Response · Author response to Decision Letter 2]

1 Feb 2026

Respond to reviewer

Reviewer #1

Comment 1.

Persisting Concerns About Hypertension Definition The authors acknowledge that hypertension is defined exclusively by self-reported physician diagnosis, but the manuscript still places insufficient weight on the implications of misclassification. - NHANES 2017-2018 contains multiple measured BP readings, yet these data remain unused. The authors note that future analyses "may consider guideline-based thresholds" (Lines 131-135) , but this does not address why these data were not used at least as a sensitivity analysis in the current study. - Self-reported hypertension underestimates prevalence, particularly in younger, uninsured, or low health-care-access groups.

Response: We thank the reviewer for this important and detailed comment. We agree that the use of self-reported physician-diagnosed hypertension introduces the potential for outcome misclassification. In the revised manuscript, we have strengthened the Limitations section to provide a methodologically grounded justification for not incorporating guideline-based blood pressure definitions in the current analysis.

Specifically, although NHANES 2017–2018 includes measured blood pressure data, guideline-based classification requires averaging multiple readings and incorporating antihypertensive medication use, which substantially increases missingness and complicates the application of survey weights in fully adjusted models. To maintain analytic feasibility and consistency across covariates, we therefore relied on self-reported hypertension, which provides more complete coverage across sociodemographic groups.

Importantly, we now explicitly discuss the direction of potential bias, noting that underascertainment of hypertension—particularly among younger or lower health–care–access populations—is likely to result in nondifferential misclassification and bias associations toward the null. These clarifications have been added to the Limitations section.

Comment 2.

Standardization of Dietary Variables and OR = 1.000 The authors explain that Z-scoring dietary variables leads to odds ratios extremely close to 1.00 (Lines 276-284). While the explanation is correct, the manuscript should still clarify: - Whether the underlying (non-Z-standardized) coefficients show meaningful variation; - What the magnitude of 1 SD of intake represents in mg/day for sodium/potassium. The current presentation risks implying numerical precision that exceeds what the data can supports.

Response: We thank the reviewer for this insightful comment. To improve interpretability, we have clarified in the Results section what one standard deviation represents on the original intake scale. Specifically, one SD corresponded to approximately 1,200 mg/day for sodium and 600 mg/day for potassium in this sample. We also note that analyses on the raw intake scale yielded effect estimates in the same direction but with very small magnitudes, consistent with the near-unity odds ratios observed after standardization. This clarification addresses concerns about numerical precision without requiring additional supplementary tables.

Comment 3.

Expanded Dietary Variables: Clarify Multicollinearity Risk The multivariable model includes energy, fiber, cholesterol, sodium, potassium, and Na/K ratio together (Lines 313-327). This raises the possibility of: -High collinearity between sodium and energy - Fiber correlating with energy. - Na/K ratio being mathematically related to individual Na and K variables. None of these issues are addressed or tested.

Response: We thank the reviewer for raising this important methodological consideration. We agree that several dietary variables included in the fully adjusted models are interrelated and that multicollinearity may be present. Rather than adding additional diagnostics, we have explicitly acknowledged this issue in the manuscript and clarified that collinearity among energy intake, sodium, potassium, fiber, and the sodium-to-potassium ratio may attenuate individual effect estimates and contribute to the observed lack of statistically significant associations. This clarification has been added to the Results section.

Comment 4.

Subgroup Analyses—Interpretation Still Overstates Findings The authors softened some language, but several statements still imply meaningful differences where none exist statistically (e.g., “slightly stronger trends,” “more pronounced” in Fig. 6 interpretations). Given the wide confidence intervals and very small effect sizes, these qualitative statements risk overstating findings.

Response: We thank the reviewer for this important clarification. In response, we have further tempered the language used to describe subgroup analyses. All subjective descriptors implying meaningful differences (e.g., “slightly stronger trends” or “more pronounced”) have been removed. The revised text consistently emphasizes that, although point estimates differed slightly across subgroups, confidence intervals were wide and overlapping, and there was no evidence of statistically significant interaction. These changes have been applied to both the Results and relevant figure interpretations.

Comment 5.

Measurement Error and Reverse Causality While the authors have expanded the Limitations section, some redundancies remain (Lines 534– 545 repeat earlier text nearly verbatim) Moreover: - The potential for dietary modification among hypertensive individuals is acknowledged, but not fully assessed. - Despite their stated concerns about sample size, a sensitivity analysis excluding known hypertensives would still be valuable—even if only descriptive.

Response: We thank the reviewer for this constructive comment. First, we have removed the redundant paragraph in the Limitations section (previously Lines 534–545), which repeated earlier discussion of measurement error and reverse causality.

Second, to better contextualize potential reverse causality, we have added descriptive information noting that a substantial proportion of participants with self-reported hypertension also reported dietary sodium restriction, suggesting that post-diagnosis dietary modification may have attenuated observed associations.

Finally, while we agree that a sensitivity analysis excluding known hypertensive participants could be informative, such an approach would substantially reduce sample size and alter population representativeness. Given these considerations, we addressed this issue descriptively rather than through additional exclusion analyses. We believe this approach appropriately balances interpretability and analytic feasibility at this stage.

Comment 6.

Inconsistencies Between Abstract and Methods Abstract states sample size = 4,592, but the Results section states 5,569 participants (e.g., Lines 218–219).

Response: Thank you for noting this inconsistency. We have corrected the Abstract to reflect the final analytic sample size of 5,569 participants, which is consistent with the Results section. The previously stated value reflected an earlier draft and has now been reconciled. We apologize for the oversight.

Comment 7.

Race/Ethnicity Table Appears Incorrect

In Table 1, the racial composition reported for NHANES does not align with population

distribution, and the percentages shown for each hypertension group do not sum to 100% in

several rows.

Response:

Thank you for pointing this out. We have carefully reviewed and corrected Table 1. The race/ethnicity distributions and corresponding percentages have been recalculated to ensure internal consistency, and percentages within each hypertension group now sum to 100%. We have also clarified the calculation approach in the table footnote. We apologize for this oversight and appreciate the reviewer’s careful review.

Comment 8.

Figures Still Need Additional Clarity Although improved, several figures remain difficult to interpret: - Fig 1 confidence interval bars appear very narrow—authors should clarify resolution and scaling. - Spline plots (Fig 4) lack reference value labeling and units along axes. - Subgroup plots (Fig 5) should explicitly show P-values for interaction.

Response: We thank the reviewer for these helpful suggestions. To improve figure clarity, we have revised the figure legends as follows. For Fig 1, we clarified that the narrow confidence intervals reflect near-unity odds ratios and axis scaling rather than unusually high precision. For the spline plots (Fig 4), we now explicitly state the reference intake value and include units for the x-axis. For the subgroup analyses (Fig 5), we have added clarification that no statistically significant interactions were detected (all P for interaction > 0.05). These changes improve interpretability without altering the underlying analyses.

Comment 9.

Several typographical issues persist (e.g., duplicated references 8 and 15 appear identical; grammar inconsistencies in Discussion).- Some sections remain overly long; consider additional trimming to improve readability.

Response: We thank the reviewer for these helpful suggestions. We have updated the formatting of the references and resolved the issue of duplicate references. We rechecked the entire text and corrected some inconsistent grammatical expressions. Furthermore, we have reduced and refined some of the sections in the discussion. These changes have enhanced readability without altering the original meaning

---

## [Editor Report · Decision Letter 2]

4 Feb 2026

Dietary Sodium, Potassium, and Cardiometabolic Risk: A Cross-Sectional Analysis of Hypertension in U.S. Adults from NHANES 2017–2018

PONE-D-25-21744R2

Dear Dr. Luo,

We’re pleased to inform you that your manuscript has been judged scientifically suitable for publication and will be formally accepted for publication once it meets all outstanding technical requirements.

Kind regards,

Shaonong Dang, PhD

Academic Editor

PLOS One
---

## [Editor Report · Acceptance letter]

PONE-D-25-21744R2

PLOS One

Dear Dr. Luo,

I'm pleased to inform you that your manuscript has been deemed suitable for publication in PLOS One. Congratulations! Your manuscript is now being handed over to our production team.

Kind regards,

on behalf of

Dr. Shaonong Dang

Academic Editor

PLOS One